# Assessing Negative Welfare Measures for Wild Invertebrates: The Case for Octopuses

**DOI:** 10.3390/ani13193021

**Published:** 2023-09-26

**Authors:** Michaella P. Andrade, Charles Morphy D. Santos, Mizziara M. M. De Paiva, Sylvia L. S. Medeiros, C. E. O’Brien, Françoise D. Lima, Janaina F. Machado, Tatiana S. Leite

**Affiliations:** 1Graduate Program in Evolution and Diversity, Federal University of ABC, Av. dos Estados, 5001, Bairro Bangu, Santo André 09210-580, Brazil; charlesmorphy@gmail.com; 2Graduate Program in Neurosciences, Brain Institute, Federal University of Rio Grande do Norte, Natal 59078-900, Brazil; mizziaradepaiva@gmail.com (M.M.M.D.P.); sylvia.lsmedeiros@gmail.com (S.L.S.M.); 3The School for Field Studies Center for Marine Resource Studies, Cockburn Harbour TKCA 1ZZ, Turks and Caicos Islands; cobrien@fieldstudies.org; 4OKEANOS, Institute of Marine Sciences, University of the Azores, 9901862 Horta, Portugal; limad.francoise@gmail.com; 5Regional Program for Development and Environment, Federal University of Rio Grande do Norte, Natal 59078-900, Brazil; machadojanainaf@gmail.com; 6Department of Ecology and Zoology, Federal University of Santa Catarina, Florianópolis 88040-900, Brazil; tati.polvo@gmail.com

**Keywords:** body patterns, Cephalopoda, field study, sentience

## Abstract

**Simple Summary:**

Wild octopuses face constant challenges to their survival, such as sublethal predation and conflicts with conspecifics, and can also be affected by human activity (e.g., interactions with divers or with fishing gear) and natural disturbances (e.g., storms, heat waves). These events can impact octopus welfare, especially when they induce stress, facilitate disease transmission, or result in starvation. Understanding the natural behavior of octopuses in the wild can help determine parameters that can be used to assess octopuses’ health and welfare. Here, we use photographic and video records of wild octopuses in a variety of negative contexts, with the goal of identifying any potential metrics of welfare. We compare these observations with published data on octopus welfare in captivity and postulate potential consequences of decreased welfare for wild octopuses. Six measures of negative welfare identified for captive animals occurred in wild octopuses as well. We also identified two new measures of negative welfare unique to wild octopuses. This study identified the first set of criteria that can be used to non-invasively assess octopus welfare in the wild. We encourage further development of non-lethal and minimally invasive techniques to quantify welfare in wild invertebrates.

**Abstract:**

Welfare metrics have been established for octopuses in the laboratory, but not for octopuses living in the wild. Wild octopuses are constantly exposed to potentially harmful situations, and the ability to assess the welfare status of wild octopuses could provide pertinent information about individuals’ health and species’ resilience to stressors. Here, we used underwater photos and videos to identify injuries and stress-related behaviors in wild *Octopus insularis* in a variety of contexts, including interacting with fishermen, interacting with other octopuses and fish, proximity to predators, in den, foraging, and in senescence. We adapted established metrics of octopus welfare from the laboratory to these wild octopuses. In addition to observing all of the stress measures, we also identified two previously unknown measures associated with decreased welfare: (1) a half white eye flash and (2) a half-and-half blotch body pattern. More than half of the individuals analyzed had arm loss, and almost half of the individuals had skin injuries. We also observed that irregular chromatophore expression and abnormal motor coordination were associated with interactions with fishermen. This is the first study to apply measures of welfare from the laboratory to wild octopuses. Our results may also aid in the identification of welfare measures for other wild invertebrates.

## 1. Introduction

Animal welfare can be defined as the quality of life experienced by sentient animals [1]. Because octopuses and other cephalopods are animals with highly developed central nervous systems and advanced cognitive abilities, many researchers argue that they are sentient, experiencing physical, mental, and emotional states [2,3] and avoid contexts in which they experience pain [4].

Shallow-water octopuses are generally semelparous with a single reproductive period at the end of their lives. Females produce hundreds to thousands of eggs, of which only a few individuals survive to adulthood [5,6]. As soft-bodied secondary consumers, octopuses are subject to predation, both lethal and sub-lethal (arm loss), from larger bony fish, elasmobranchs, marine mammals, and seabirds [7]. Wild octopuses must constantly weigh the benefit of foraging against the cost of defense and the threat of predation [8] and interactions with other animals [9,10] and must select suitable shelters and perform regular maintenance on this den [11]. Lethal and sublethal predation, fishing, and agonistic and interspecific interactions are some of the daily challenges that can affect the welfare of octopuses [12,13,14], resulting in physical damage, stress, and/or pain [4].

Standardized measures to quantify overall welfare can be affected by factors including living conditions, diet, and access to enrichment [15]. Measures based on changes in physiology and behavior are used to assess the welfare of vertebrates, but relatively few welfare measures for invertebrates have been validated thus far [16,17]. Differences in physiology, anatomy, and behavior preclude the use of the same welfare measures for vertebrates and invertebrates [16]. Moreover, most research is limited to animal welfare in laboratories and captivity, with the welfare of wildlife having received much less attention [18,19,20].

The literature describing the association between body patterns and behavioral repertoires in octopuses is still scarce [21]. Most works focus on analyzing body patterns related to camouflage [22], changes in chromatophores [23], and systematics [24]. There is a gap in understanding how much body patterns and their different components can be related to these animals’ welfare and emotional states.

Human activities such as hunting and fishing have a detrimental effect on the welfare of many wild animals, resulting in injuries, stress, and pain [18,25]. Although stress and pain can increase fitness in the short term, over time, they can negatively impact animal welfare and health ([19] *contra*, see [15]). The impact of physical injuries on octopus mortality remains poorly understood. Data regarding the effects of injuries and the expression of abnormal behaviors in octopuses offer crucial information about the challenges they face in the wild [19,26,27,28].

Octopus fishing is a common practice in many coastal communities [29]. In Brazil, octopuses are commonly caught by using a “bicheiro” (an improvised harpoon) [30] but discarded or used as prey for fishing if the individual is too small to sell (Leite, personal report). In fish, the practice of “catch and release” induces, in addition to physical injuries, the release of stress hormones and changes in behavior [31]. The hormonal and behavioral consequences of “catch-and-release” in octopuses should be explored in the coming years, especially with increasing evidence that these animals experience pain, distress, suffering, and lasting harm [4,32,33,34,35].

Our study focuses on wild *Octopus insularis* (Leite and Haimovici 2008), a medium-sized benthic species found in the tropical and subtropical southwestern Atlantic. This species has been studied for the past 25 years in Brazil by Projeto Cephalopoda, which has led to the creation of an extensive database of photographs, videos, and scientific information [36,37,38]. Here, we evaluate photographic and video evidence of injuries and corresponding behavioral changes in wild *O. insularis*. We then compare these data with reports of injuries and concomitant behaviors observed in captive octopuses to isolate potential welfare measures for wild octopuses. In total, we found eight measures related to negative welfare in wild *O. insularis*, six already identified in captive octopuses and two unique to wild animals. This is the first study to apply welfare measures used for captive octopuses to wild ones. Our results are valuable first steps for future in situ investigations seeking to assess welfare and identify the least harmful means of human interaction with wild octopuses.

## 2. Materials and Methods

Field observations and recording occurred from 2005 to 2023, generating a database of 674 photos and 411 videos of *O. insularis* in various contexts, as well as of octopus fishermen at work. The fieldwork was carried out in several areas of Brazil: Fernando de Noronha National Park; the São Pedro and São Paulo and Abrolhos Archipelagos; the Atol das Rocas Biological Reserve; Nisia Floresta; Maceió; and Salvador. The regions sampled are tropical with a humid coastal climate and high air temperatures throughout the year, ranging between 20 °C and 28 °C. Observational sessions utilized ad libitum and focal follow methods [39] with SCUBA and snorkel. Photographs donated by other researchers, artisanal fishermen, and marine photographers were also utilized.

From the database, we selected media based on photo and video quality; the proportion of the animal’s body that was visible; the presence of visible injuries; and for a larger context, the presence of behaviors that could be related to welfare [40]. We collected the following metadata from each photo or video: location, collection date, photographer, number of photographs/videos per animal, behavioral context, number, location and type of skin injuries, number of missing arms, body pattern, presence of irregular chromatophore expression, and abnormal body position. We collected the following data from both photographs and videos: (1) body patterns; (2) type of injuries; (3) number of injuries; (4) irregular chromatophore expression; and (5) abnormal body position. For all variables, except for the number of injuries, we used a binary system to indicate presence, absence, or inability to visualize. In the case of the number of injuries, we quantified both the total number of injuries and the number of missing arms. When analyzing body patterns, we considered the occurrence (presence/absence) of each body pattern in videos, as a single video could contain multiple body patterns. In contrast, for photos, only one body pattern per photo was recorded given the limitation of capturing multiple patterns in a single image.

In describing body patterns and components, we utilized existing cephalopod nomenclature whenever possible [41] (see Appendix A for definitions of body patterns). Injuries, context, and behaviors suggestive of impaired welfare present in the database are defined in Table 1 (results). To identify behavioral context, we recorded environmental characteristics and information about the octopuses from the media sources and data collectors’ notes.

From the eight welfare measures discussed here, we chose four for statistical analysis as they were the most commonly observed in our database (Table 2). The following factors were used as categorical variables: (1) body pattern, (2) number of injuries, (3) irregular chromatophore expression, and (4) abnormal body position. To analyze the association between the contexts and these variables, we used chi-square tests [42] in RStudio version 4.2.1. [43]. We set an alpha of *p* < 0.05, and Bonferroni adjustments were utilized for multiple comparisons [44].

## 3. Results

### 3.1. Welfare Measures and Contexts

We found that the total sample was 57 wild octopuses. Of these, 44 octopuses were recorded in 175 photographs, and 13 octopuses were recorded in 23 videos. Our data came from seven different areas in Brazil: 51% from Fernando de Noronha National Park (n = 29), 19% from Nisia Floresta (Rio Grande do Norte) (n = 11), 16% from Atol das Rocas Biological Reserve (n = 9), 7% from Bahia (n = 4), 4% from Macéio (Alagoas) (n = 2), 2% from Abrolhos Archipelago (n = 1), and 2% from São Pedro and São Paulo Archipelago (n = 1).

Eight indicators of welfare were identified in these photos and videos (Table 1). In addition, seven different contexts that could potentially impact welfare were identified: (1) interacting with fishermen (in the act of fishing) (Figure 1); (2) agonistic interactions with other octopuses, either at a distance or via physical contact; (3) interactions with fish; (4) proximity to a predator (when a predator was close by or physically interacting with the octopus); (5) in den (when the octopus was inside a sheltered crevice); (6) foraging (when the octopus was moving along the substrate or swimming in the water column looking for food); and (7) entering a state of senescence (near the end of its lifespan). The highest percentage of observations were from interactions with fishermen (42.11%; n = 24), foraging (22.81%; n = 13), and interactions with fish (14.04%, n = 8) (Table 2).

The most commonly utilized potential welfare measure was arm loss (Table 3, Figure 2): 38 individuals (67% of the total sample) were missing at least one arm. Skin injuries were also common, with 24 individuals (42% of the total sample) possessing them. Injuries occurred on a variety of body parts, including the arms, mantle, head, eyes, suckers, and arm web, with some individuals possessing injuries in multiple locations. Irregular chromatophore expression was observed in 15 individuals, 12 of them during interactions with fishermen. There was a significant relationship between injury location and context (χ^2^ = 68.64; gl = 42; *p*-value= 0.007). Specifically, during agonistic interactions with conspecifics, more injuries were observed on the mantle, while for individuals in den, more injuries were observed on the head and eyes (z-value = −3.32). Irregular chromatophore expression also varied by context (χ^2^ = 48.86; df = 12; *p*-value < 0.0001), more often associated with interactions with fishermen (z-value = −3.04). Abnormal body positions also varied by context (χ^2^ = 49.7; df = 12; *p*-value= 0.000001577), most associated with interactions with fishermen (z-value = −3.04).

We identified 82 occurrences within our observations of 57 octopuses in which body patterns could be discerned (Table 4). Approximately half of the patterns were blotched and mottled. Other body patterns observed were uniform dark, uniform light, different variations of longitudinal stripes, half-and-half blotch, and deimatic displays.

The body patterns observed in our samples (Table 4) were significantly affected by each of the contexts considered here (χ^2^ = 101.77; gl = 54; *p*-value < 0.0001). Our data demonstrated a significant positive relationship between the blotch body pattern and interactions with fishermen, the half-and-half blotch pattern and interactions with fish, and the uniform light pattern and senescence (z-value= −3.38) (Figure 3).

### 3.2. Notes on Octopus Behavior

We noted some interesting behaviors in a few of the individuals we observed in the field. At Boldró Beach (Fernando de Noronha National Park), an octopus was observed with prolonged papillae expression, excessive dermal mucus, and fixed irregular spots of dark chromatophores on the right side of the mantle (Figure 4A–C,E). Right arm I, left arm I, and right arm III were lost, with differing amounts of regenerating tissue. The octopus exhibited a half-and-half body pattern, with the dark side corresponding with the wounded side of the mantle (Figure 4C). We also observed episodes of “white flash” around its pupils, which was sometimes only expressed on the lower part of the above pupil (half white eye flash).

We also recorded a senescent female at Cacimba do Padre Beach (Fernando de Noronha National Park) with pale skin (uniform light) and a near-total loss of muscle tone, as demonstrated by disordered movement of the arms. This female had likely gone through a long period of fasting during the maturation of her eggs (Figure 4T). This is the first record of senescence in a wild female *O. insularis* to date.

At the Atol das Rocas Biological Reserve, we video-recorded an agonistic interaction between two males. The posture of one of the combatants was similar to the “Nosferatu posture” previously reported for *Octopus tetricus* Gould 1852 [45], in which the octopus displays a uniform dark body pattern, with extended arm membranes and a mantle elevated above the rest of the body, moving via backward jetting. We were able to observe the octopuses engaged in “full attack” mode, in which one individual wrapped its arms around the other, and both were thrown against the substrate.

The “startle response” behavior was seen in eight octopuses in response to territorial fish (mostly in Fernando de Noronha National Park, Nisia Floresta, and Atol das Rocas Biological Reserve). When octopuses were approached by fish, either making physical contact or by rapidly accelerating toward the octopus, they exhibited a half-and-half blotch body pattern, with well-spots on the mantle directed toward the fish (Figure 4P).

Overall, we identified two previously undescribed behaviors in wild octopuses: half-and-half blotch (a startle response) and white eye flash in the lower part of the pupil (derivation of the “eye flash” described in [9]). We also observed the half-and-half body pattern, previously reported only in sleep phases [46], which was expressed when octopuses were awake.

### 3.3. Comparisons of Welfare Measures between Captive and Wild Octopuses

Based on a review of the literature covering cephalopod welfare in captivity, we compiled a set of potential measures for gauging welfare in wild octopuses (Table 5).

We also observed two possible measures not reported before in captive octopuses: (1) the half-and-half blotch body pattern, which seems to indicate that the octopus is experiencing acute stress, and (2) the half white eye flash, associated with excitement and acute stress.

## 4. Discussion

This is the first study to apply welfare metrics used in captivity to wild octopuses. We identified these metrics using photographs and videos of wild octopuses, an approach already used for other species such as horses [19], whales [52], and penguins [53]. Our study reiterates the efficacy of minimally invasive assessments in the study and pursuit of animal welfare.

*Octopus insularis*, a diurnal species [54], predominantly inhabits shallow, well-lit waters characterized by high visibility, often exceeding 10 m [6]. These conditions facilitate behavioral studies by allowing individuals to be tracked closely and continuously using cameras placed in front of their dens or in focal follows during excursions from the den [54]. Most of the media we collected were recorded by or during interactions with fishermen, reflecting the fact that *O. insularis* is a highly fished species in Brazil [14,29] and that these animals regularly interact with fishermen in many Brazilian coastal areas. Nevertheless, to date, no study has attempted to understand the impacts of capture and handling by fishermen on free-living octopuses.

We show that all six metrics of negative welfare measures identified for captive octopuses [13,40,47,48,49,50,51] also occur in wild ones. In particular, injuries to the body are quite common. In captive settings, mantle lesions in *Eledone cirrhosa* Lamarck, 1798 have been largely attributed to octopuses “jetting” against tank walls [55]. In the wild, interactions with fishermen and predators and agonistic interactions with conspecifics have the potential to cause injuries. In our sampling, bodily injuries were seen most often in interactions with fishermen and when the octopus was in its den. This may be a protective behavior in response to discomfort, pain [4], or increased vulnerability. Another possibility is that animals with this type of injury need to sleep longer to conserve energy and aid healing, like vertebrates [56].

Arm loss is common in octopuses [13,26,27], especially through predation. Reports have shown a tendency to lose the first left arm (LI) in *O. bimaculatus* Verrill, 1883; *O. bimaculoides* Pickford and McConnaughey, 1949; and *O. rubescens* Berry 1953 [13]. As octopuses use their arms to manipulate prey and objects and to obtain sensory information from the environment [3], arm loss impacts everyday behaviors. For example, arm loss in wild *Abdopus sp.* Norman and Finn 2001 impacts male competition and mating success [57]. Arm loss has been used for welfare assessment in captive octopuses: the loss of three or more arms is the most worrying situation, requiring immediate action (sometimes euthanasia) [40]. In our data, the loss of only one arm was much more frequent than the loss of two, but about a quarter of the animals had lost three or more arms. We observed a tendency to lose arms LI, R1, and R2, suggesting that these arms were used most often in risky behaviors [13,58] or for defense against attacks. Arm loss can have negative implications for the welfare and survival of octopuses in the wild, increasing challenges in foraging and handling food, as well as leading to potential disadvantages in competing with other individuals.

Irregular chromatophore expression was observed in 50% of the interactions between *O. insularis* and fishermen. In addition to skin injuries, the rough handling of these animals can damage their chromatophores. Our field observations strongly suggest that a blow from a “bicheiro” hook can cause chromatophore failure, resulting in areas that are permanently dark or light. Defective chromatophore animals in the wild are probably at a higher risk of predation due to disruption to their camouflage [59].

One of the octopuses displayed a buildup of mucus on its skin, potentially indicative of bacterial growth [27,48]. Normally, an octopus removes excess dermal mucus via grooming, and in laboratory settings, octopuses that cease or reduce grooming are those that show other indicators of poor health [47]. Although our sampling was restricted to a single individual, excessive dermal mucus in wild octopuses has potential consequences for welfare, including an increased likelihood of parasite accumulation on the skin, which can result in chronic stress and even death.

In a captive environment, the excessive expression of papillae on the mantle and above the eyes may indicate excited states in octopuses [40,50]. The exposure of papillae in wild octopuses is related to camouflage and is usually expressed for only short periods of time. We observed an octopus that expressed its papillae for more than 1 min. Although more research is required, this is likely another indicator of poor welfare.

We found a strong association between abnormal motor coordination and interactions with fishermen but not between the octopuses’ abnormal motor coordination and foraging. It is worth noting that, during fishing activities, octopuses are often struck by fishing gear, such as “bicheiros”, repeatedly without being collected. Octopuses that are particularly degraded may be used as bait for other fishing activities or discarded. The discarded individuals often exhibit anomalous coordination, which is otherwise observed only after egg laying and during senescence [51]. Sometimes, the animal’s impaired behavior resembles an elegant “dance” and is recorded and shared on social media [60]. This behavior may be an indication of pain or discomfort. During this abnormal behavior, the muscle tone of the arms visibly decreases, and the animal moves almost exclusively at the whim of the current. This abnormal motor coordination compromises welfare, as it negatively affects the ability to perform basic maintenance behaviors such as grooming, which may lead to the accumulation of parasites on the skin. In the laboratory, abnormal motor behavior is often associated with imminent death [61].

We found two other potential measures of welfare that have not been noted from captive observations: the half-and-half blotch body pattern and the half white eye flash. The blotch components of the half-and-half blotch pattern are characterized by bright white leucophores. Asymmetric body patterns occur in other cephalopods such as cuttlefish during stressful situations like predator approach and during intraspecific competition for mates [62]. The asymmetric behavior that we call half-and-half blotch was expressed in response to territorial fish, which also elicited startle reactions. According to the literature [8,9,62,63,64], when fish are present around foraging octopuses, this can have both negative and positive effects on their fitness. The negative effects include competition for food [62] and the possibility of alerting predators to their location. However, in at least one instance, the interaction is known to be collaborative [65]. Here, during the half-and-half blotch body pattern, the octopus displayed its blotch pattern toward the approaching territorial fish and the brown or mottle pattern on the other side of the body. A similar situation has also been reported in cuttlefish in potentially stressful situations, such as predator approach and interspecific interactions [62], and in reef squid, where unilateral components appear when fish approach [66]. This is the first description of a unilateral pattern in response to an abrupt approach or touch by a fish in an octopus. Further studies are necessary to determine if the half-and-half blotch body pattern is an acute stress response in all octopuses.

A white flash around the eyes has been previously reported as a response to approaching territorial fish and other disturbances when an octopus is in its den [67], potentially as an anti-predator defense, confusing the predator during a chase [68]. At Boldró Beach, Fernando de Noronha, we recorded a juvenile octopus away from its den (as indicated by a lack of prey remains nearby) that displayed a white flash only on the lower half of its iris (Figure 4B) in response to an approaching snorkeler. This octopus was missing its arms and had an injury on its mantle. It is possible that the white flash is a measure of acute stress in *O. insularis* or else that some injury to his body rendered him unable to shine the upper part of his white eye, but more evidence is needed to confirm this hypothesis.

The investigation of chromatic patterns as potential indicators of stress has been conducted in several captive cephalopods [69,70,71]. For instance, lighter chromatic patterns, raised papillae, and vertical stripes on the body have been found to be associated with stress in *O. bimaculoides* [65,66]. In this study, we found the uniform light body pattern (seen in the senescent female), the blotch pattern (seen during interactions with fishermen), and the half-and-half blotch pattern (in startle responses in interactions with fishes) to be associated with stress and thus potential visual indicators of stress in *O. insularis*.

Our dataset has limitations primarily arising from the challenges of studying wild octopuses non-invasively. These limitations include difficulties in capturing photos and videos, the absence of unambiguous information about the sex and size of the octopuses, and potential biases introduced by the presence of human observers during recordings.

## 5. Conclusions

We identified several potential measures of negative welfare in wild *O. insularis* of Brazil, which will enable researchers to visually assess animal conditions in the field. Our results are the first steps for future in situ investigations that require assessments of the welfare of wild octopuses.

The impact of various stressors on the reproductive success and population trends of octopuses remains unstudied. Detailed and systematic knowledge of the factors that affect welfare status in wild octopuses is not only essential for understanding how these potentially sentient animals experience the world but also allows us to predict how different anthropogenic stressors in the marine environment (e.g., pollution, negative interactions with tourism, and fishing) will impact them.

## Figures and Tables

**Figure 1 animals-13-03021-f001:**
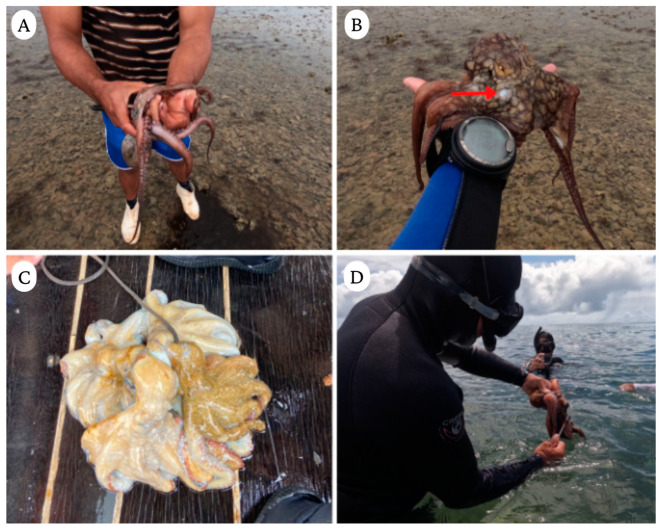
*Octopus insularis* fishery in Brazil (Bahia). (**A**) An octopus collected by a fisherman; (**B**) an octopus with a skin injury (red arrow) possibly due to a prior capture; (**C**) octopuses being stored after collection by fishermen; and (**D**) an octopus collected with a “bicheiro”. Photos by Lucas Ribeiro.

**Figure 2 animals-13-03021-f002:**
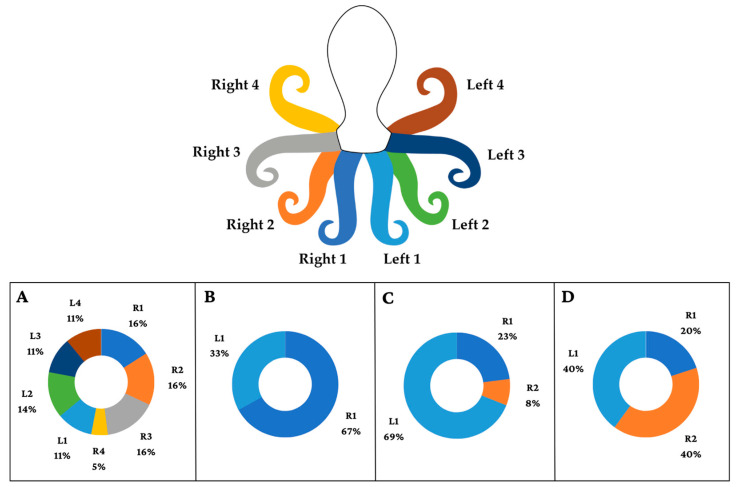
Illustration showing the percentage with which each arm is lost (R1, R2, R3, R4, L1, L2, L3, and L4) and percentage of losses in different contexts: (**A**) interactions with fishermen; (**B**) interactions with fish; (**C**) foraging; and (**D**) interactions with other octopuses. Some individuals had more than one missing arm. The contexts of “in den”, “entering a state of senescence”, and “proximity to predator” were not included since arm loss was rarely observed in these contexts.

**Figure 3 animals-13-03021-f003:**
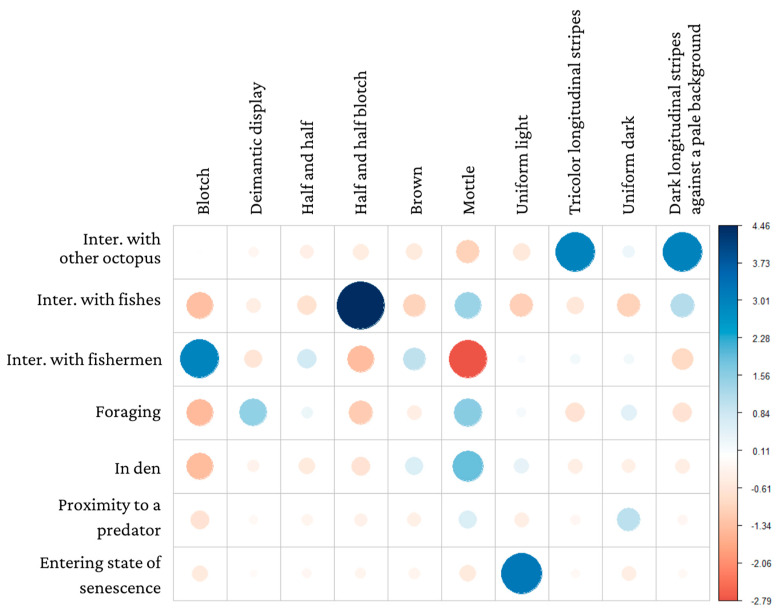
Correlation plot for chi-squared test residuals for each context against body pattern/component (χ^2^ = 101.77; gl = 54; *p*-value < 0.0001, 0.00009164). Positive associations are in blue, and negative associations are in red.

**Figure 4 animals-13-03021-f004:**
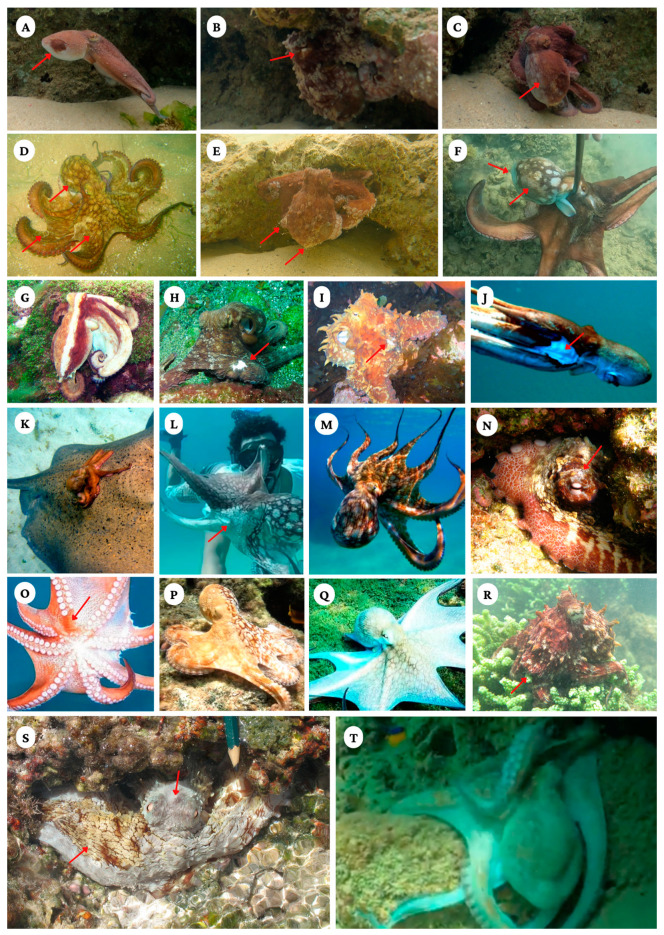
Octopuses under both natural and anthropogenic stressors. (**A**) Dark brown lesion on the right side of the mantle (arrow) indicating the site of a previous injury; (**B**) half white eye flash, pale eye flash on the lower part of the pupil (arrow); (**C**) half-and-half pattern; (**D**) octopus with one arm missing, scratches on the skin (arrows) and expressing a blotch body pattern; (**E**) octopus with uniform dark pattern, loss of arms, and excess dermal mucus (arrows); (**F**) octopus after being caught with a “bicheiro”, showing irregular chromatophore expression (arrows); (**G**) tricolor longitudinal stripes; (**H**) octopus with mottle pattern and wound on one arm (arrow); (**I**) octopus with mottle pattern, raised papillae, and wound on one arm (arrow); (**J**) octopus with an injury on its arms and mantle (arrow); (**K**) an injured octopus lying on a stingray (*Dasyatis americana* Hildebrand and Schroeder, 1928); (**L**) gisherman catching octopus with visible irregular chromatophore expression (arrow); (**M**) octopus after being hit with the fisherman’s “bicheiro”; (**N**) lesion above the eye (arrow), white above the pupil, and mottle pattern; (**O**) injury that resulted in the loss of suckers (arrow); (**P**) startle response at an approaching territorial fish, half-and-half blotch body pattern (2/4 s); (**Q**) octopus expressing a deimatic pattern; (**R**) octopus foraging over algae with one missing arm (arrow); (**S**) octopus in den with irregular chromatophore expression: a small portion of the body is expressing the mottle pattern, while the greater portion expresses a pale pattern—normally, these areas would express the same body pattern; and (**T**) senescent female. Photos by S. Medeiros (**A**–**C**); Buia (**D**); M. Paiva (**E**); L. Ribeiro (**F**); Drauzio (**G**,**P**); T. Leite (**H**,**I**,**R**); Cosme Johnny (**J**); A.T. Souza (**K**); Nego Noronha (**L**,**M**,**O**,**Q**); F. Lima (**N**); H. Bouth (**S**); and Gibson Lemos (**T**).

**Table 1 animals-13-03021-t001:** Potential welfare measures in wild *O. insularis*, including injuries, aberrant behaviors or body positions, irregularities in chromatophore expression, and definitions of each negative welfare measure.

Welfare Measures	Definitions	Contexts
Arm loss	Absence of a part of or entire arm(s) or presence of regenerated arm tissue.	Interacting with fishermen, fish, and other octopuses and in den and foraging
Skin injuries	Fixed, irregular, pale, or reddish spots or scars. Often near the eyes/head, arms (excluding whole or partial arm loss), arm web, suckers, or mantle.	Interacting with fishermen, fish, and other octopuses; proximity to a predator; in den and foraging
Irregular chromatophore expression	Uncoordinated color changes or areas of the body with an unchanging uniform color or pattern.	Interacting with fishermen, in den, and foraging
Abnormal body position in water column	Abnormal arm movement and/or loss of muscle tone (e.g., arms).	Interacting with fishermen and entering a state of senescence
Half-and-half blotch	Display of two different body patterns with a distinct longitudinal line dividing them. One half of the body shows the blotch pattern, consisting of white circles on a brown background. The other half of the body shows mottled or brown patterns. The texture is smooth. This pattern can occur when the animal is still or in motion.	Interactions with fish
Excess dermal mucus *	Presence of excess dermal mucus on the mantle and/or parts of the arms, presumably due to a lack of grooming activity (cleaning maneuver).	Foraging
Excess papillae expression *	Papillae are raised for an extended period (at least 1 min).	Foraging
White eye flash *	The area immediately around the pupil flashes pale, usually for two seconds or more. Paleness may occur on both upper and lower halves of the iris or only the lower half.	Foraging

* Reported in a small number (n < 2) of records.

**Table 2 animals-13-03021-t002:** The number and percentage of individual *O. insularis* observed in seven contexts potentially impacting welfare.

Context	% Data	Number of Individuals
Interactions with fishermen	42.11%	24
Foraging	22.81%	13
Interactions with fishes	14.04%	8
In den	12.28%	7
Agonistic interactions with other octopus	3.51%	2
Proximity to predator	3.51%	2
Entering a state of senescence	1.75%	1
Total	100%	57

**Table 3 animals-13-03021-t003:** Number of times each potential welfare indicator was utilized according to context.

	Number of Instances						
Welfare Measures	Interactions with Fishermen	Foraging	Interactions with Fish	In Den	Interactions with Other Octopuses	Proximity to Predator	Entering a State of Senescence
Arm loss	21	11	2	2	2	-	-
Skin injuries	13	3	1	4	2	1	-
Irregular chromatophore expression	12	2	0	1	0	0	-
Abnormal body position	9	0	0	0	0	0	1
Total occurrences	55	16	16	7	4	1	1

**Table 4 animals-13-03021-t004:** Summary of body patterns and components observed in our database (“-” indicates absence).

Body Patterns and Components	Interactions with Fishermen	Foraging	Interactions with Fish	In Den	Interactions with Other Octopuses	Proximity to Predator	Entering a State of Senescence
Blotch	16	2	1	-	1	-	-
Mottle	1	10	7	5	-	1	-
Uniform dark	6	3	1	1	1	1	-
Uniform light	3	2	-	1	-	-	1
Light brown	3	1	-	1	-	-	-
Deimatic display	-	1	-	-	-	-	-
Tricolor longitudinal stripes	1	-	-	-	1	-	-
White longitudinal stripe	-	-	1	-	1	-	-
Half and half	2	1	-	-	-	-	-
Half-and-half blotch	-	-	5	-	-	-	-
Total occurrences	32	20	15	8	4	2	1

**Table 5 animals-13-03021-t005:** Injuries and behaviors described for different octopus species in the laboratory and applicable to wild *Octopus insularis*, along with their putative consequences.

Welfare Measure	Species	Consequences for Wild Octopuses	References
Skin injuries	General for Cephalopoda	Problems with camouflage; increased detection by predators; viral or bacterial diseases; inflammation and hyperalgesia; acute or chronic stress; death.	[47,48,49]
	*Octopus vulgaris*, *O. joubini*, *O. briareus*		
Arm loss	General for Cephalopoda	Problems in foraging, locomotion, defense, male competition, and mating success; difficulties in manipulating objects and prey; physical costs beyond energy demands for arm regeneration; death.	[13,47]
	*O. bimaculatus*, *O. bimaculoides*, *O. rubescens*, *Abdopus* sp.		
Chromatophore failure	General for Cephalopoda	Problems with camouflage; increased predator detection; death.	[47]
Excess dermal mucus	General for Cephalopoda	Parasite accumulation; chronic stress; death.	[47,48]
Extended papillae expression	*O. vulgaris*,	Acute or chronic stress	[40,50]
	*Enteroctopus dofleini*		
Abnormal body position	*O. bimaculoides*	Chronic stress; problems in foraging, locomotion, and defense; difficulties in manipulating objects and prey; parasite accumulation; death.	[47,51]

## Data Availability

Photos and videos are unavailable due to privacy restrictions. They are part of the Projeto Cephalopda scientific database.

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
