# Peer review of "Assessing Negative Welfare Measures for Wild Invertebrates: The Case for Octopuses"

_animals, 2023, doi:10.3390/ani13193021_

Round 1

Reviewer 1 Report

This is a well written and considered submission in an area which remains understudied. I don't have a lot of comments (which is unusual for me).

Can the intro perhaps explain the literature about negative affect in invertebrates (esp octopus) especially the importance of patternation and why "blotching" as an indicator? If there is no published info on this it would be useful for there to be some kind of supposition. I note this is in the discussion but would be of value in the intro.

Can you provide a little more about the methods applied to the videos as compared to the photographs? How were the points/behaviours of interest applied, given that it was able to be recorded as an instantaneous, continuous or present/absent factor? Obviously behaviour would be hard to record from a photograph.

L310 "a blow"

Author Response

Thank you very much for taking the time to review this manuscript.
Please find the detailed responses below and the corresponding revisions.

Unfortunately, the literature that seeks to understand these associations between body patterns and welfare is underdeveloped. Regarding this subject, we added the following paragraph to the introduction:

"The literature describing the association between body patterns and behavioral repertoires in octopuses is still scarce [21]. Most of the works focus on analyzing body patterns related to camouflage [22], changes in chromatophores [23], and systematics [24]. There is a gap in understanding how much body patterns and their different components can be related to these animals' welfare and emotional states."

The following references were also added:

[21] Packard, A., & Hochberg, F.G. Skin patterning in Octopus and other genera. In Symp. Zool. Soc. Lond 1977, 38, pp. 191-231.

[22] Guidetti, G., Levy, G., Matzeu, G., Finkelstein, J.M., Levin, M., & Omenetto, F.G. (2021). Unmixing octopus camouflage by multispectral mapping of Octopus bimaculoides' chromatic elements. Nanophotonics 2021, 10(9), 2441-2450. https://doi.org/10.1515/nanoph-2021-0102 

[23] Messenger, J.B. (2001). Cephalopod chromatophores: neurobiology and natural history. Biological Reviews 2001, 76(4), 473-528. DOI: 10.1017/s1464793101005772

[24] Huffard, C.L. Ethogram of Abdopus aculeatus (d'Orbigny, 1834) (Cephalopoda: Octopodidae): Can behavioural characters inform octopodid taxomony and systematics?. Journal of Molluscan Studies 2007, 73(2), 185-193. https://doi.org/10.1093/mollus/eym015 

We included the following paragraph in the Methods section:

"We collected the following data for both photographs and videos: (1) body patterns; (2) type of injuries; (3) number of injuries; (4) irregular chromatophore expression; and (5) abnormal body position. For all variables, except for the number of injuries, we used a binary system to indicate presence, absence, or inability to visualize. In the case of the number of injuries, we quantified both the total number of injuries and the number of missing arms. When analyzing body patterns, we considered the occurrence (presence/absence) of each body pattern in videos, as a single video could contain multiple body patterns. In contrast, for photos, only one body pattern per photo was recorded due to the limitation of capturing multiple patterns in a single image."

L310: Correction made

Reviewer 2 Report

This study evaluated a database of photographs and videos collected in Brazil by Projeto Cephalopoda from 2005 to 2023 for injuries and corresponding behavioural changes in wild octopus (Octopus insularis) in a variety of contexts. These contexts include interacting with fishermen, interacting with other octopuses and fishes, proximity to predators, in den, foraging, and in senescence. The data obtained were compared to reports of injuries and concomitant behaviours observed in captive octopuses to identify potential welfare measures for wild octopuses.

Comments

This manuscript is clear and detailed. Materials and methods are thoroughly described. There is a logical progression from identifying the eight welfare measures with their contexts and utilisation (Tables 1, 2 and 3) to a summary of body patterns (Table 4) and a correlation of body patterns with context (Figure 3). The manuscript is well illustrated (Figures 1, 2 and 4).

Figure 2 shows arm loss occurs most commonly in the first pair and there is an uneven distribution between the right and left arm within this pair. The discussion (lines 303-304) suggests the pattern of arm loss indicates which arms are used most often in risky behaviours and in defence against attack. These data may also imply handedness, as demonstrated in other animals.

The results showed six negative welfare measures identified for captive octopuses also occur in wild Octopus insularis. Two new negative welfare measures identified in wild octopuses were half white eye flash and a half-and-half blotch body pattern. This manuscript thus provides a robust set of non-invasive criteria that can be used to assess octopus welfare in the wild.

Minor comments

Page 3 Section 3.1 lines 138 and 139 contains this ‘We identified 57 wild octopuses in 175 photographs and 23 videos. 44 octopuses were recorded in photos and 13 in videos.’ It may be clearer if this was changed to ‘The total sample was 57 wild octopuses. Of these, 44 octopuses were recorded in 175 photographs and 13 octopuses were recorded in 23 videos.’

Page 4 Line 152. Change ‘recordings’ to ‘observations’ because recordings implies video.

Page 11 Line 310 ‘ablow’ should be ‘a blow’

The quality and editing of English language are excellent. Some minor comments were provided to improve clarity.

Author Response

Thank you very much for taking the time to review this manuscript.
Please find the detailed responses below and the corresponding revisions. 

Lines 146 and 147 - We changed according to reviewer comment - Suggestion accepted.

We identified the total sample was 57 wild octopuses. Of these, 44 octopuses were rec-orded in 175 photographs, and 13 octopuses were recorded in 23 videos.”

Line 160 - We changed according to reviewer comment - Suggestion accepted.

“observations”

- ‘ablow’ should be ‘a blow’ - Ok, correction made.

We also modified the acknowledgments to include thanks for reviewers' considerations - Line 431 and 432 - "We would like to thank the reviewers for their relevant considerations to improve the work."